# Neural Architecture Search
# with Bayesian Optimisation and Optimal Transport

**Kirthevasan Kandasamy, Willie Neiswanger, Jeff Schneider, Barnabás Póczos, Eric P Xing**

Carnegie Mellon University,     Petuum Inc.

{kandasamy, willie, schneide, bapoczos, epxing}@cs.cmu.edu

## Abstract

Bayesian Optimisation (BO) refers to a class of methods for global optimisation of a function $f$ which is only accessible via point evaluations. It is typically used in settings where $f$ is expensive to evaluate. A common use case for BO in machine learning is model selection, where it is not possible to analytically model the generalisation performance of a statistical model, and we resort to noisy and expensive training and validation procedures to choose the best model. Conventional BO methods have focused on Euclidean and categorical domains, which, in the context of model selection, only permits tuning scalar hyper-parameters of machine learning algorithms. However, with the surge of interest in deep learning, there is an increasing demand to tune neural network *architectures*. In this work, we develop NASBOT, a Gaussian process based BO framework for neural architecture search. To accomplish this, we develop a distance metric in the space of neural network architectures which can be computed efficiently via an optimal transport program. This distance might be of independent interest to the deep learning community as it may find applications outside of BO. We demonstrate that NASBOT outperforms other alternatives for architecture search in several cross validation based model selection tasks on multi-layer perceptrons and convolutional neural networks.

## 1    Introduction

In many real world problems, we are required to sequentially evaluate a noisy black-box function $f$ with the goal of finding its optimum in some domain $\mathcal{X}$. Typically, each evaluation is expensive in such applications, and we need to keep the number of evaluations to a minimum. Bayesian optimisation (BO) refers to an approach for global optimisation that is popularly used in such settings. It uses Bayesian models for $f$ to infer function values at unexplored regions and guide the selection of points for future evaluations. BO has been successfully applied for many optimisation problems in optimal policy search, industrial design, and scientific experimentation. That said, the quintessential use case for BO in machine learning is *model selection* [14, 40]. For instance, consider selecting the regularisation parameter $\lambda$ and kernel bandwidth $h$ for an SVM. We can set this up as a zeroth order optimisation problem where our domain is a two dimensional space of $(\lambda, h)$ values, and each function evaluation trains the SVM on a training set, and computes the accuracy on a validation set. The goal is to find the model, i.e. hyper-parameters, with the highest validation accuracy.

The majority of the BO literature has focused on settings where the domain $\mathcal{X}$ is either Euclidean or categorical. This suffices for many tasks, such as the SVM example above. However, with recent successes in deep learning, neural networks are increasingly becoming the method of choice for many machine learning applications. A number of recent work have designed novel neural network architectures to significantly outperform the previous state of the art [12, 13, 37, 45]. This motivates studying model selection over the space of neural architectures to optimise for generalisation performance. A critical challenge in this endeavour is that evaluating a network via train and validation procedures is very expensive. This paper proposes a BO framework for this problem.

While there are several approaches to BO, those based on Gaussian processes (GP) [35] are most common in the BO literature. In its most unadorned form, a BO algorithm operates sequentially, starting at time 0 with a GP prior for $f$; at time $t$, it incorporates results of evaluations from $1, \ldots, t-1$ in the form of a posterior for $f$. It then uses this posterior to construct an acquisition function $\varphi_t$, where $\varphi_t(x)$ is a measure of the value of evaluating $f$ at $x$ at time $t$ if our goal is to maximise $f$. Accordingly, it chooses to evaluate $f$ at the maximiser of the acquisition, i.e. $x_t = \mathrm{argmax}_{x \in \mathcal{X}} \varphi_t(x)$. There are two key ingredients to realising this plan for GP based BO. First, we need to quantify the similarity between two points $x, x'$ in the domain in the form of a kernel $\kappa(x, x')$. The kernel is needed to define the GP, which allows us to reason about an unevaluated value $f(x')$ when we have already evaluated $f(x)$. Secondly, we need a method to maximise $\varphi_t$.

These two steps are fairly straightforward in conventional domains. For example, in Euclidean spaces, we can use one of many popular kernels such as Gaussian, Laplacian, or Matérn; we can maximise $\varphi_t$ via off the shelf branch-and-bound or gradient based methods. However, when each $x \in \mathcal{X}$ is a neural network architecture, this is not the case. Hence, our challenges in this work are two-fold. First, we need to *quantify (dis)similarity between two networks*. Intuitively, in Fig. 1, network 1a is more similar to network 1b, than it is to 1c. Secondly, we need to be able to traverse the space of such networks to *optimise the acquisition function*. Our main contributions are as follows.

1. We develop a (pseudo-)distance for neural network architectures called OTMANN (Optimal Transport Metrics for Architectures of Neural Networks) that can be computed efficiently via an optimal transport program.
2. We develop a BO framework for optimising functions on neural network architectures called NASBOT (Neural Architecture Search with Bayesian Optimisation and Optimal Transport). This includes an evolutionary algorithm to optimise the acquisition function.
3. Empirically, we demonstrate that NASBOT outperforms other baselines on model selection tasks for multi-layer perceptrons (MLP) and convolutional neural networks (CNN). Our python implementations of OTMANN and NASBOT are available at `github.com/kirthevasank/nasbot`.

**Related Work:** Recently, there has been a surge of interest in methods for neural architecture search [1, 6, 8, 21, 25, 26, 30, 32, 36, 41, 51–54]. We discuss them in detail in the Appendix due to space constraints. Broadly, they fall into two categories, based on either evolutionary algorithms (EA) or reinforcement learning (RL). EA provide a simple mechanism to explore the space of architectures by making a sequence of changes to networks that have already been evaluated. However, as we will discuss later, they are not ideally suited for optimising functions that are expensive to evaluate. While RL methods have seen recent success, architecture search is in essence an *optimisation* problem – find the network with the lowest validation error. There is no explicit need to maintain a notion of state and solve credit assignment [43]. Since RL is a fundamentally more difficult problem than optimisation [16], these approaches need to try a very large number of architectures to find the optimum. This is not desirable, especially in computationally constrained settings.

None of the above methods have been designed with a focus on the expense of evaluating a neural network, with an emphasis on being judicious in selecting which architecture to try next. Bayesian optimisation (BO) uses introspective Bayesian models to carefully determine future evaluations and is well suited for expensive evaluations. BO usually consumes more computation to determine future points than other methods, but this pays dividends when the evaluations are very expensive. While there has been some work on BO for architecture search [2, 15, 28, 40, 44], they have only been applied to optimise feed forward structures, e.g. Fig. 1a, but not Figs. 1b, 1c. We compare NASBOT to one such method and demonstrate that feed forward structures are inadequate for many problems.

## 2 Set Up

Our goal is to maximise a function $f$ defined on a space $\mathcal{X}$ of neural network architectures. When we evaluate $f$ at $x \in \mathcal{X}$, we obtain a possibly noisy observation $y$ of $f(x)$. In the context of architecture search, $f$ is the performance on a validation set after $x$ is trained on the training set. If $x_\star = \mathrm{argmax}_{\mathcal{X}} f(x)$ is the optimal architecture, and $x_t$ is the architecture evaluated at time $t$, we want $f(x_\star) - \max_{t \leq n} f(x_t)$ to vanish fast as the number of evaluations $n \to \infty$. We begin with a review of BO and then present a graph theoretic formalism for neural network architectures.

### 2.1 A brief review of Gaussian Process based Bayesian Optimisation

A GP is a random process defined on some domain $\mathcal{X}$, and is characterised by a mean function $\mu : \mathcal{X} \to \mathbb{R}$ and a (covariance) kernel $\kappa : \mathcal{X}^2 \to \mathbb{R}$. Given $n$ observations $\mathcal{D}_n = \{(x_i, y_i)\}_{i=1}^n$, where

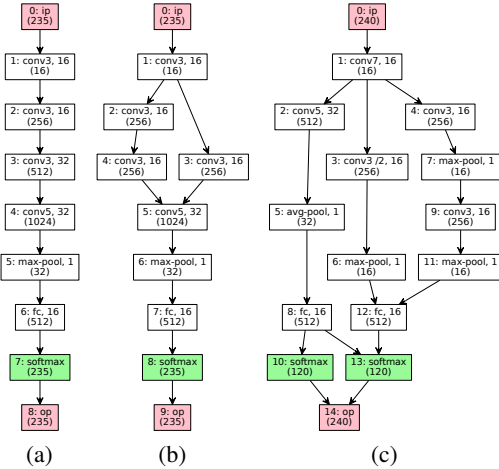

Figure 1: An illustration of some CNN architectures. In each layer, $i$: indexes the layer, followed by the label (e.g `conv3`), and then the number of units (e.g. number of filters). The input and output layers are pink while the decision (`softmax`) layers are green.

*From Section 3:* The layer mass is denoted in parentheses. The following are the normalised and unnormalised distances $d, \bar{d}$. All self distances are 0, i.e. $d(\mathcal{G},\mathcal{G}) = \bar{d}(\mathcal{G},\mathcal{G}) = 0$. Unnormalised: $d(a,b) = 175.1$, $d(a,c) = 1479.3$, $d(b,c) = 1621.4$. Normalised: $\bar{d}(a,b) = 0.0286$, $\bar{d}(a,c) = 0.2395$, $\bar{d}(b,c) = 0.2625$.

$x_i \in \mathcal{X}, y_i = f(x_i) + \epsilon_i \in \mathbb{R}$, and $\epsilon_i \sim \mathcal{N}(0, \eta^2)$, the posterior process $f|\mathcal{D}_n$ is also a GP with mean $\mu_n$ and covariance $\kappa_n$. Denote $Y \in \mathbb{R}^n$ with $Y_i = y_i$, $k, k' \in \mathbb{R}^n$ with $k_i = \kappa(x, x_i)$, $k_i' = \kappa(x', x_i)$, and $K \in \mathbb{R}^{n \times n}$ with $K_{i,j} = \kappa(x_i, x_j)$. Then, $\mu_n, \kappa_n$ can be computed via,

$$\mu_n(x) = k^\top (K + \eta^2 I)^{-1} Y, \qquad \kappa_n(x, x') = \kappa(x, x') - k^\top (K + \eta^2 I)^{-1} k'. \qquad (1)$$

For more background on GPs, we refer readers to Rasmussen and Williams [35]. When tasked with optimising a function $f$ over a domain $\mathcal{X}$, BO models $f$ as a sample from a GP. At time $t$, we have already evaluated $f$ at points $\{x_i\}_{i=1}^{t-1}$ and obtained observations $\{y_i\}_{i=1}^{t-1}$. To determine the next point for evaluation $x_t$, we first use the posterior GP to define an *acquisition function* $\varphi_t : \mathcal{X} \to \mathbb{R}$, which measures the utility of evaluating $f$ at any $x \in \mathcal{X}$ according to the posterior. We then maximise the acquisition $x_t = \operatorname{argmax}_{\mathcal{X}} \varphi_t(x)$, and evaluate $f$ at $x_t$. The expected improvement acquisition [31],

$$\varphi_t(x) = \mathbb{E}\big[\max\{0, f(x) - \tau_{t-1}\} \big| \{(x_i, y_i)\}_{i=1}^{t-1}\big], \qquad (2)$$

measures the expected improvement over the current maximum value according to the posterior GP. Here $\tau_{t-1} = \operatorname{argmax}_{i < t-1} f(x_i)$ denotes the current best value. This expectation can be computed in closed form for GPs. We use EI in this work, but the ideas apply just as well to other acquisitions [3].

**GP/BO in the context of architecture search:** Intuitively, $\kappa(x, x')$ is a measure of similarity between $x$ and $x'$. If $\kappa(x, x')$ is large, then $f(x)$ and $f(x')$ are highly correlated. Hence, the GP effectively imposes a smoothness condition on $f : \mathcal{X} \to \mathbb{R}$; i.e. since networks a and b in Fig. 1 are similar, they are likely to have similar cross validation performance. In BO, when selecting the next point, we balance between *exploitation*, choosing points that we believe will have high $f$ value, and *exploration*, choosing points that we do not know much about so that we do not get stuck at a bad optimum. For example, if we have already evaluated $f(a)$, then exploration incentivises us to choose c over b since we can reasonably gauge $f(b)$ from $f(a)$. On the other hand, if $f(a)$ has high value, then exploitation incentivises choosing b, as it is more likely to be the optimum than c.

## 2.2 A Mathematical Formalism for Neural Networks

Our formalism will view a neural network as a graph whose vertices are the layers of the network. We will use the CNNs in Fig. 1 to illustrate the concepts. A neural network $\mathcal{G} = (\mathcal{L}, \mathcal{E})$ is defined by a set of layers $\mathcal{L}$ and directed edges $\mathcal{E}$. An edge $(u, v) \in \mathcal{E}$ is a ordered pair of layers. In Fig. 1, the layers are depicted by rectangles and the edges by arrows. A layer $u \in \mathcal{L}$ is equipped with a layer label $\ell\ell(u)$ which denotes the type of operations performed at the layer. For instance, in Fig. 1a, $\ell\ell(1) = \texttt{conv3}$, $\ell\ell(5) = \texttt{max-pool}$ denote a $3 \times 3$ convolution and a max-pooling operation. The attribute $\ell u$ denotes the number of computational units in a layer. In Fig. 1b, $\ell u(5) = 32$ and $\ell u(7) = 16$ are the number of convolutional filters and fully connected nodes.

In addition, each network has *decision layers* which are used to obtain the predictions of the network. For a classification task, the decision layers perform `softmax` operations and output the probabilities an input datum belongs to each class. For regression, the decision layers perform `linear` combinations of the outputs of the previous layers and output a single scalar. All networks

have at least one decision layer. When a network has multiple decision layers, we average the output of each decision layer to obtain the final output. The decision layers are shown in green in Fig. 1. Finally, every network has a unique *input layer* $u_{\mathrm{ip}}$ and *output layer* $u_{\mathrm{op}}$ with labels $\ell\ell(u_{\mathrm{ip}}) = \texttt{ip}$ and $\ell\ell(u_{\mathrm{op}}) = \texttt{op}$. It is instructive to think of the role of $u_{\mathrm{ip}}$ as feeding a data point to the network and the role of $u_{\mathrm{op}}$ as averaging the results of the decision layers. The input and output layers are shown in pink in Fig. 1. We refer to all layers that are not input, output or decision layers as *processing layers*.

The directed edges are to be interpreted as follows. The output of each layer is fed to each of its children; so both layers 2 and 3 in Fig. 1b take the output of layer 1 as input. When a layer has multiple parents, the inputs are concatenated; so layer 5 sees an input of $16 + 16$ filtered channels coming in from layers 3 and 4. Finally, we mention that neural networks are also characterised by the values of the weights/parameters between layers. In architecture search, we typically do not consider these weights. Instead, an algorithm will (somewhat ideally) assume access to an optimisation oracle that can minimise the loss function on the training set and find the optimal weights.

We next describe a distance $d : \mathcal{X}^2 \to \mathbb{R}_+$ for neural architectures. Recall that our eventual goal is a kernel for the GP; given a distance $d$, we will aim for $\kappa(x, x') = e^{-\beta d(x,x')^p}$, where $\beta, p \in \mathbb{R}_+$, as the kernel. Many popular kernels take this form. For e.g. when $\mathcal{X} \subset \mathbb{R}^n$ and $d$ is the $L^2$ norm, $p = 1, 2$ correspond to the Laplacian and Gaussian kernels respectively.

## 3   The OTMANN  Distance

To motivate this distance, note that the performance of a neural network is determined by the amount of computation at each layer, the types of these operations, and how the layers are connected. A meaningful distance should account for these factors. To that end, OTMANN is defined as the minimum of a matching scheme which attempts to match the computation at the layers of one network to the layers of the other. We incur penalties for matching layers with different types of operations or those at structurally different positions. We will find a matching that minimises these penalties, and the total penalty at the minimum will give rise to a distance. We first describe two concepts, layer masses and path lengths, which we will use to define OTMANN.

**Layer masses:** The layer masses $\ell m : \mathcal{L} \to \mathbb{R}_+$ will be the quantity that we match between the layers of two networks when comparing them. $\ell m(u)$ quantifies the significance of layer $u$. For processing layers, $\ell m(u)$ will represent the amount of computation carried out by layer $u$ and is computed via the product of $\ell u(u)$ and the number of incoming units. For example, in Fig. 1b, $\ell m(5) = 32 \times (16 + 16)$ as there are 16 filtered channels each coming from layers 3 and 4 respectively. As there is no computation at the input and output layers, we cannot define the layer mass directly as we did for the processing layers. Therefore, we use $\ell m(u_{\mathrm{ip}}) = \ell m(u_{\mathrm{op}}) = \zeta \sum_{u \in \mathcal{PL}} \ell m(u)$ where $\mathcal{PL}$ denotes the set of processing layers, and $\zeta \in (0, 1)$ is a parameter to be determined. Intuitively, we are using an amount of mass that is proportional to the amount of computation in the processing layers. Similarly, the decision layers occupy a significant role in the architecture as they directly influence the output. While there is computation being performed at these layers, this might be problem dependent – there is more computation performed at the softmax layer in a 10 class classification problem than in a 2 class problem. Furthermore, we found that setting the layer mass for decisions layers based on computation underestimates their contribution to the network. Following the same intuition as we did for the input/output layers, we assign an amount of mass proportional to the mass in the processing layers. Since the outputs of the decision layers are averaged, we distribute the mass among all decision layers; that is, if $\mathcal{DL}$ are decision layers, $\forall\, u \in \mathcal{DL}, \ell m(u) = \frac{\zeta}{|\mathcal{DL}|} \sum_{u \in \mathcal{PL}} \ell m(u)$. In all our experiments, we use $\zeta = 0.1$. In Fig. 1, the layer masses for each layer are shown in parantheses.

**Path lengths from/to** $u_{\mathrm{ip}}/u_{\mathrm{op}}$**:** In a neural network $\mathcal{G}$, a path from $u$ to $v$ is a sequence of layers $u_1, \ldots, u_s$ where $u_1 = u$, $u_s = v$ and $(u_i, u_{i+1}) \in \mathcal{E}$ for all $i \leq s - 1$. The length of this path is the number of hops from one node to another in order to get from $u$ to $v$. For example, in Fig. 1c, $(2, 5, 8, 13)$ is a path from layer 2 to 13 of length 3. Let the shortest (longest) path length from $u$ to $v$ be the smallest (largest) number of hops from one node to another among all paths from $u$ to $v$. Additionally, define the random walk path length as the expected number of hops to get from $u$ to $v$, if, from any layer we hop to one of its children chosen uniformly at random. For example, in Fig. 1c, the shortest, longest and random walk path lengths from layer 1 to layer 14 are 5, 7, and 5.67 respectively. For any $u \in \mathcal{L}$, let $\delta_{\mathrm{op}}^{\mathrm{sp}}(u), \delta_{\mathrm{op}}^{\mathrm{lp}}(u), \delta_{\mathrm{op}}^{\mathrm{rw}}(u)$ denote the length of the shortest, longest and random walk paths from $u$ to the output $u_{\mathrm{op}}$. Similarly, let $\delta_{\mathrm{ip}}^{\mathrm{sp}}(u), \delta_{\mathrm{ip}}^{\mathrm{lp}}(u), \delta_{\mathrm{ip}}^{\mathrm{rw}}(u)$ denote the corresponding lengths

|          | conv3 | conv5 | max-pool | avg-pool | fc |
|----------|-------|-------|----------|----------|-----|
| conv3    | 0     | 0.2   | $\infty$ | $\infty$ | $\infty$ |
| conv5    | 0.2   | 0     | $\infty$ | $\infty$ | $\infty$ |
| max-pool | $\infty$ | $\infty$ | 0     | 0.25     | $\infty$ |
| avg-pool | $\infty$ | $\infty$ | 0.25  | 0        | $\infty$ |
| fc       | $\infty$ | $\infty$ | $\infty$ | $\infty$ | 0 |

Table 1: An example label mismatch cost matrix $M$. There is zero cost for matching identical layers, $< 1$ cost for similar layers, and infinite cost for disparate layers.

for walks from the input $u_{\text{ip}}$ to $u$. As the layers of a neural network can be topologically ordered[1], the above path lengths are well defined and finite. Further, for any $s \in \{\text{sp,lp,rw}\}$ and $t \in \{\text{ip,op}\}$, $\delta_t^s(u)$ can be computed for all $u \in \mathcal{L}$, in $\mathcal{O}(|\mathcal{E}|)$ time (see Appendix A.3 for details).

We are now ready to describe OTMANN. Given two networks $\mathcal{G}_1 = (\mathcal{L}_1, \mathcal{E}_1), \mathcal{G}_2 = (\mathcal{L}_2, \mathcal{E}_2)$ with $n_1, n_2$ layers respectively, we will attempt to match the layer masses in both networks. We let $Z \in \mathbb{R}_+^{n_1 \times n_2}$ be such that $Z(i,j)$ denotes the amount of mass matched between layer $i \in \mathcal{G}_1$ and $j \in \mathcal{G}_2$. The OTMANN distance is computed by solving the following optimisation problem.

$$\underset{Z}{\text{minimise}} \quad \phi_{\text{lmm}}(Z) + \phi_{\text{nas}}(Z) + \nu_{\text{str}}\phi_{\text{str}}(Z) \qquad (3)$$

$$\text{subject to} \sum_{j \in \mathcal{L}_2} Z_{ij} \leq \ell m(i), \sum_{i \in \mathcal{L}_1} Z_{ij} \leq \ell m(j), \ \forall i,j$$

The label mismatch term $\phi_{\text{lmm}}$, penalises matching masses that have different labels, while the structural term $\phi_{\text{str}}$ penalises matching masses at structurally different positions with respect to each other. If we choose not to match any mass in either network, we incur a non-assignment penalty $\phi_{\text{nas}}$. $\nu_{\text{str}} > 0$ determines the trade-off between the structural and other terms. The inequality constraints ensure that we do not over assign the masses in a layer. We now describe $\phi_{\text{lmm}}, \phi_{\text{nas}}$, and $\phi_{\text{str}}$.

*Label mismatch penalty* $\phi_{\text{lmm}}$: We begin with a label penalty matrix $M \in \mathbb{R}^{L \times L}$ where $L$ is the number of all label types and $M(\mathtt{x}, \mathtt{y})$ denotes the penalty for transporting a unit mass from a layer with label $\mathtt{x}$ to a layer with label $\mathtt{y}$. We then construct a matrix $C_{\text{lmm}} \in \mathbb{R}^{n_1 \times n_2}$ with $C_{\text{lmm}}(i,j) = M(\ell\ell(i), \ell\ell(j))$ corresponding to the mislabel cost for matching unit mass from each layer $i \in \mathcal{L}_1$ to each layer $j \in \mathcal{L}_2$. We then set $\phi_{\text{lmm}}(Z) = \langle Z, C_{\text{lmm}} \rangle = \sum_{i \in \mathcal{L}_1, j \in \mathcal{L}_2} Z(i,j)C(i,j)$ to be the sum of all matchings from $\mathcal{L}_1$ to $\mathcal{L}_2$ weighted by the label penalty terms. This matrix $M$, illustrated in Table 1, is a parameter that needs to be specified for OTMANN. They can be specified with an intuitive understanding of the functionality of the layers; e.g. many values in $M$ are $\infty$, while for similar layers, we choose a value less than 1.

*Non-assignment penalty* $\phi_{\text{nas}}$: We set this to be the amount of mass that is unassigned in both networks, i.e. $\phi_{\text{nas}}(Z) = \sum_{i \in \mathcal{L}_1} \left(\ell m(i) - \sum_{j \in \mathcal{L}_2} Z_{ij}\right) + \sum_{j \in \mathcal{L}_2} \left(\ell m(j) - \sum_{i \in \mathcal{L}_1} Z_{ij}\right)$. This essentially implies that the cost for not assigning unit mass is 1. The costs in Table 1 are defined relative to this. For similar layers $\mathtt{x}, \mathtt{y}$, $M(\mathtt{x}, \mathtt{y}) \ll 1$ and for disparate layers $M(\mathtt{x}, \mathtt{y}) \gg 1$. That is, we would rather match conv3 to conv5 than not assign it, provided the structural penalty for doing so is small; conversely, we would rather not assign a conv3, than assign it to fc. This also explains why we did not use a trade-off parameter like $\nu_{\text{str}}$ for $\phi_{\text{lmm}}$ and $\phi_{\text{nas}}$ – it is simple to specify reasonable values for $M(\mathtt{x}, \mathtt{y})$ from an understanding of their functionality.

*Structural penalty* $\phi_{\text{str}}$: We define a matrix $C_{\text{str}} \in \mathbb{R}^{n_1 \times n_2}$ where $C_{\text{str}}(i,j)$ is small if layers $i \in \mathcal{L}_1$ and $j \in \mathcal{L}_2$ are at structurally similar positions in their respective networks. We then set $\phi_{\text{str}}(Z) = \langle Z, C_{\text{str}} \rangle$. For $i \in \mathcal{L}_1, \ j \in \mathcal{L}_2$, we let $C_{\text{str}}(i,j) = \frac{1}{6} \sum_{s \in \{\text{sp, lp, rw}\}} \sum_{t \in \{\text{ip,op}\}} |\delta_t^s(i) - \delta_t^s(j)|$ be the average of all path length differences, where $\delta_t^s$ are the path lengths defined previously. We define $\phi_{\text{str}}$ in terms of the shortest/longest/random-walk path lengths from/to the input/output, because they capture various notions of information flow in a neural network; a layer's input is influenced by the paths the data takes before reaching the layer and its output influences all layers it passes through before reaching the decision layers. If the path lengths are similar for two layers, they are likely to be at similar structural positions. Further, this form allows us to solve (3) efficiently via an OT program and prove distance properties about the solution. If we need to compute pairwise distances for several networks, as is the case in BO, the path lengths can be pre-computed in $\mathcal{O}(|\mathcal{E}|)$ time, and used to construct $C_{\text{str}}$ for two networks at the moment of computing the distance between them.

This completes the description of our matching program. In Appendix A, we prove that (3) can be formulated as an Optimal Transport (OT) program [47]. OT is a well studied problem with several efficient solvers [33]. Our theorem below, shows that the solution of (3) is a distance.

| Operation | Description |
|---|---|
| dec_single | Pick a layer at random and decrease the number of units by $1/8$. |
| dec_en_masse | Pick several layers at random and decrease the number of units by $1/8$ for all of them. |
| inc_single | Pick a layer at random and increase the number of units by $1/8$. |
| inc_en_masse | Pick several layers at random and increase the number of units by $1/8$ for all of them. |
| dup_path | Pick a random path $u_1, \ldots, u_k$, duplicate $u_2, \ldots, u_{k-1}$ and connect them to $u_1$ and $u_k$. |
| remove_layer | Pick a layer at random and remove it. Connect the layer's parents to its children if necessary. |
| skip | Randomly pick layers $u, v$ where $u$ is topologically before $v$. Add $(u, v)$ to $\mathcal{E}$. |
| swap_label | Randomly pick a layer and change its label. |
| wedge_layer | Randomly remove an edge $(u, v)$ from $\mathcal{E}$. Create a new layer $w$ and add $(u, w), (w, v)$ to $\mathcal{E}$. |

Table 2: Descriptions of modifiers to transform one network to another. The first four change the number of units in the layers but do not change the architecture, while the last five change the architecture.

**Theorem 1.** *Let $d(\mathcal{G}_1, \mathcal{G}_2)$ be the solution of (3) for networks $\mathcal{G}_1, \mathcal{G}_2$. Under mild regularity conditions on $M$, $d(\cdot, \cdot)$ is a pseudo-distance. That is, for all networks $\mathcal{G}_1, \mathcal{G}_2, \mathcal{G}_3$, it satisfies, $d(\mathcal{G}_1, \mathcal{G}_2) \geq 0$, $d(\mathcal{G}_1, \mathcal{G}_2) = d(\mathcal{G}_2, \mathcal{G}_1)$, $d(\mathcal{G}_1, \mathcal{G}_1) = 0$ and $d(\mathcal{G}_1, \mathcal{G}_3) \leq d(\mathcal{G}_1, \mathcal{G}_2) + d(\mathcal{G}_2, \mathcal{G}_3)$.*

For what follows, define $\bar{d}(\mathcal{G}_1, \mathcal{G}_2) = d(\mathcal{G}_1, \mathcal{G}_2)/(tm(\mathcal{G}_1) + tm(\mathcal{G}_2))$ where $tm(\mathcal{G}_i) = \sum_{u \in \mathcal{L}_i} \ell m(u)$ is the total mass of a network. Note that $\bar{d} \leq 1$. While $\bar{d}$ does not satisfy the triangle inequality, it provides a useful measure of dissimilarity normalised by the amount of computation. Our experience suggests that $d$ puts more emphasis on the amount of computation at the layers over structure and vice versa for $\bar{d}$. Therefore, it is prudent to combine both quantities in any downstream application. The caption in Fig. 1 gives $d, \bar{d}$ values for the examples in that figure when $\nu_{\mathrm{str}} = 0.5$.

We conclude this section with a couple of remarks. First, OTMANN shares similarities with Wasserstein (earth mover's) distances which also have an OT formulation. However, it is not a Wasserstein distance itself—in particular, the supports of the masses and the cost matrices change depending on the two networks being compared. Second, while there has been prior work for defining various distances and kernels on graphs, we cannot use them in BO because neural networks have additional complex properties in addition to graphical structure, such as the type of operations performed at each layer, the number of neurons, etc. The above work either define the distance/kernel between vertices or assume the same vertex (layer) set [9, 23, 29, 38, 49], none of which apply in our setting. While some methods do allow different vertex sets [48], they cannot handle layer masses and layer similarities. Moreover, the computation of the above distances are more expensive than OTMANN. Hence, these methods cannot be directly plugged into BO framework for architecture search.

In Appendix A, we provide additional material on OTMANN. This includes the proof of Theorem 1, a discussion on some design choices, and implementation details such as the computation of the path lengths. Moreover, we provide illustrations to demonstrate that OTMANN is a meaningful distance for architecture search. For example, a t-SNE embedding places similar architectures close to each other. Further, scatter plots showing the validation error vs distance on real datasets demonstrate that networks with small distance tend to perform similarly on the problem.

## 4 NASBOT

We now describe NASBOT, our BO algorithm for neural architecture search. Recall that in order to realise the BO scheme outlined in Section 2.1, we need to specify (a) a kernel $\kappa$ for neural architectures and (b) a method to optimise the acquisition $\varphi_t$ over these architectures. Due to space constraints, we will only describe the key ideas and defer all details to Appendix B.

As described previously, we will use a negative exponentiated distance for $\kappa$. Precisely, $\kappa = \alpha e^{-\beta d} + \bar{\alpha} d^{-\bar{\beta}\bar{d}}$, where $d, \bar{d}$ are the OTMANN distance and its normalised version. We mention that while this has the form of popular kernels, we do not know yet if it is in fact a kernel. In our experiments, we did not encounter an instance where the eigenvalues of the kernel matrix were negative. In any case, there are several methods to circumvent this issue in kernel methods [42].

We use an evolutionary algorithm (EA) approach to optimise the acquisition function (2). For this, we begin with an initial pool of networks and evaluate the acquisition $\varphi_t$ on those networks. Then we generate a set of $N_{\mathrm{mut}}$ mutations of this pool as follows. First, we stochastically select $N_{\mathrm{mut}}$ candidates from the set of networks already evaluated such that those with higher $\varphi_t$ values are more likely to be selected than those with lower values. Then we modify each candidate, to produce a new architecture. These modifications, described in Table 2, might change the architecture either by

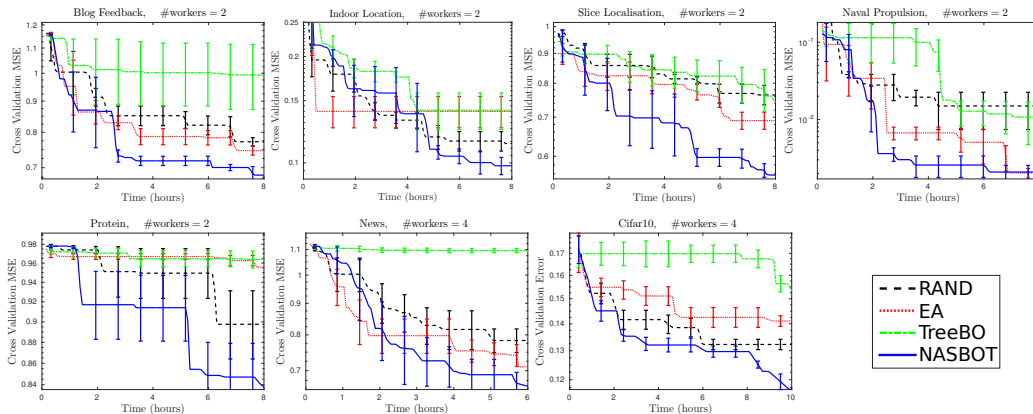

Figure 2: *Cross validation results:* In all figures, the $x$ axis is time. The $y$ axis is the mean squared error (MSE) in the first 6 figures and the classification error in the last. Lower is better in all cases. The title of each figure states the dataset and the number of parallel workers (GPUs). All figures were averaged over at least 5 independent runs of each method. Error bars indicate one standard error.

increasing or decreasing the number of computational units in a layer, by adding or deleting layers, or by changing the connectivity of existing layers. Finally, we evaluate the acquisition on this $N_{mut}$ mutations, add it to the initial pool, and repeat for the prescribed number of steps. While EA works fine for cheap functions, such as the acquisition $\varphi_t$ which is analytically available, it is not suitable when evaluations are expensive, such as training a neural network. This is because EA selects points for future evaluations that are already close to points that have been evaluated, and is hence inefficient at exploring the space. In our experiments, we compare NASBOT to the same EA scheme used to optimise the acquisition and demonstrate the former outperforms the latter.

We conclude this section by observing that this framework for NASBOT/OTMANN has additional flexibility to what has been described. If one wishes to tune over drop-out probabilities, regularisation penalties and batch normalisation at each layer, they can be treated as part of the layer label, via an augmented label penalty matrix $M$ which accounts for these considerations. If one wishes to jointly tune other scalar hyper-parameters (e.g. learning rate), they can use an existing kernel for euclidean spaces and define the GP over the joint architecture + hyper-parameter space via a product kernel. BO methods for early stopping in iterative training procedures [17–20, 22] can be easily incorporated by defining a *fidelity space*. Using a line of work in scalable GPs [39, 50], one can apply our methods to challenging problems which might require trying a very large number ($\sim$100K) of architectures. These extensions will enable deploying NASBOT in large scale settings, but are tangential to our goal of introducing a BO method for architecture search.

## 5   Experiments

**Methods:** We compare NASBOT to the following baselines. RAND: random search; EA (Evolutionary algorithm): the same EA procedure described above. TreeBO [15]: a BO method which only searches over feed forward structures. Random search is a natural baseline to compare optimisation methods. However, unlike in Euclidean spaces, there is no natural way to randomly explore the space of architectures. Our RAND implementation, operates in exactly the same way as NASBOT, except that the EA procedure is fed a random sample from $\mathrm{Unif}(0,1)$ instead of the GP acquisition each time it evaluates an architecture. Hence, RAND is effectively picking a random network from the same space explored by NASBOT; neither method has an unfair advantage because it considers a different space. While there are other methods for architecture search, their implementations are highly nontrivial and are not made available.

**Datasets:** We use the following datasets: blog feedback [4], indoor location [46], slice localisation [11], naval propulsion [5], protein tertiary structure [34], news popularity [7], Cifar10 [24]. The first six are regression problems for which we use MLPs. The last is a classification task on images for which we use CNNs. Table 3 gives the size and dimensionality of each dataset. For the first 6 datasets, we use a $0.6 - 0.2 - 0.2$ train-validation-test split and normalised the input and output to have zero mean and unit variance. Hence, a constant predictor will have a mean squared error of approximately 1. For Cifar10 we use $40K$ for training and $10K$ each for validation and testing.

| Method | Blog $(60K, 281)$ | Indoor $(21K, 529)$ | Slice $(54K, 385)$ | Naval $(12K, 17)$ | Protein $(46K, 9)$ | News $(40K, 61)$ | Cifar10 $(60K, 3K)$ | Cifar10 $150K$ iters |
|---|---|---|---|---|---|---|---|---|
| RAND | 0.780 $\pm$ 0.034 | **0.115** $\pm$**0.023** | 0.758 $\pm$ 0.041 | 0.0103 $\pm$ 0.002 | 0.948 $\pm$ 0.024 | **0.762** $\pm$**0.013** | 0.1342 $\pm$ 0.002 | 0.0914 $\pm$ 0.008 |
| EA | 0.806 $\pm$ 0.040 | 0.147 $\pm$ 0.010 | 0.733 $\pm$ 0.041 | **0.0079** $\pm$**0.004** | 1.010 $\pm$ 0.038 | **0.758** $\pm$**0.038** | 0.1411 $\pm$ 0.002 | 0.0915 $\pm$ 0.010 |
| TreeBO | 0.928 $\pm$ 0.053 | 0.168 $\pm$ 0.023 | 0.759 $\pm$ 0.079 | 0.0102 $\pm$ 0.002 | 0.998 $\pm$ 0.007 | 0.866 $\pm$ 0.085 | 0.1533 $\pm$ 0.004 | 0.1121 $\pm$ 0.004 |
| NASBOT | **0.731** $\pm$**0.029** | **0.117** $\pm$**0.008** | **0.615** $\pm$**0.044** | **0.0075** $\pm$**0.002** | **0.902** $\pm$**0.033** | **0.752** $\pm$**0.024** | **0.1209** $\pm$**0.003** | **0.0869** $\pm$**0.004** |

Table 3: The first row gives the number of samples $N$ and the dimensionality $D$ of each dataset in the form $(N, D)$. The subsequent rows show the regression MSE or classification error (lower is better) on the *test set* for each method. The last column is for Cifar10 where we took the best models found by each method in 24K iterations and trained it for $120K$ iterations. When we trained the VGG-19 architecture using our training procedure, we got test errors 0.1718 (60K iterations) and 0.1018 (150K iterations).

**Experimental Set up:** Each method is executed in an asynchronously parallel set up of 2-4 GPUs, That is, it can evaluate multiple models in parallel, with each model on a single GPU. When the evaluation of one model finishes, the methods can incorporate the result and immediately re-deploy the next job without waiting for the others to finish. For the blog, indoor, slice, naval and protein datasets we use 2 GeForce GTX 970 (4GB) GPUs and a computational budget of 8 hours for each method. For the news popularity dataset we use 4 GeForce GTX 980 (6GB) GPUs with a budget of 6 hours and for Cifar10 we use 4 K80 (12GB) GPUs with a budget of 10 hours. For the regression datasets, we train each model with stochastic gradient descent (SGD) with a fixed step size of $10^{-5}$, a batch size of 256 for 20K batch iterations. For Cifar10, we start with a step size of $10^{-2}$, and reduce it gradually. We train in batches of 32 images for 60K batch iterations. The methods evaluate between 70-120 networks depending on the size of the networks chosen and the number of GPUs.

**Results:** Fig. 2 plots the best validation score for each method against time. In Table 3, we present the results on the test set with the best model chosen on the basis of validation set performance. On the Cifar10 dataset, we also trained the best models for longer ($150K$ iterations). These results are in the last column of Table 3. We see that NASBOT is the most consistent of all methods. The average time taken by NASBOT to determine the next architecture to evaluate was 46.13s. For RAND, EA, and TreeBO this was 26.43s, 0.19s, and 7.83s respectively. The time taken to train and validate models was on the order of 10-40 minutes depending on the model size. Fig. 2 includes this time taken to determine the next point. Like many BO algorithms, while NASBOT's selection criterion is time consuming, it pays off when evaluations are expensive. In Appendices B and C, we provide additional details on the experiment set up and conduct synthetic ablation studies by holding out different components of the NASBOT framework. We also illustrate some of the best architectures found—on many datasets, common features were long skip connections and multiple decision layers.

Finally, we note that while our Cifar10 experiments fall short of the current state of the art [25, 26, 53], the amount of computation in these work is several orders of magnitude more than ours (both the computation invested to train a single model and the number of models trained). Further, they use constrained spaces specialised for CNNs, while NASBOT is deployed in a very general model space. We believe that our results can also be improved by employing enhanced training techniques such as image whitening, image flipping, drop out, etc. For example, using our training procedure on the VGG-19 architecture [37] yielded a test set error of 0.1018 after $150K$ iterations. However, VGG-19 is known to do significantly better on Cifar10. That said, we believe our results are encouraging and lay out the premise for BO for neural architectures.

## 6 Conclusion

We described NASBOT, a BO framework for neural architecture search. NASBOT finds better architectures for MLPs and CNNs more efficiently than other baselines on several datasets. A key contribution of this work is the efficiently computable OTMANN distance for neural network architectures, which may be of independent interest as it might find applications outside of BO. Our code for NASBOT and OTMANN will be made available.

**Acknolwedgements**

We would like to thank Guru Guruganesh and Dougal Sutherland for the insightful discussions. This research is partly funded by DOE grant DESC0011114, NSF grant IIS1563887, and the Darpa D3M program. KK is supported by a Facebook fellowship and a Siebel scholarship.

## Footnotes

[1]A topological ordering is an ordering of the layers $u_1, \ldots, u_{|\mathcal{L}|}$ such that $u$ comes before $v$ if $(u,v) \in \mathcal{E}$.

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
