[Supplementary Material · nasbot_full.pdf]

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

_{\text{op}}$. Similarly, let $\delta_{\text{ip}}^{\text{sp}}(u), \delta_{\text{ip}}^{\text{lp}}(u), \delta_{\text{ip}}^{\text{rw}}(u)$ denote the corresponding lengths

|          | conv3 | conv5 | max-pool | avg-pool | fc |
|----------|-------|-------|----------|----------|-----|
| conv3    | 0     | 0.2   | $\infty$ | $\infty$ | $\infty$ |
| conv5    | 0.2   | 0     | $\infty$ | $\infty$ | $\infty$ |
| max-pool | $\infty$ | $\infty$ | 0 | 0.25 | $\infty$ |
| avg-pool | $\infty$ | $\infty$ | 0.25 | 0 | $\infty$ |
| fc       | $\infty$ | $\infty$ | $\infty$ | $\infty$ | 0 |

Table 1: An example label mismatch cost matrix $M$. There is zero cost for matching identical layers, $< 1$ cost for similar layers, and infinite cost for disparate layers.

for walks from the input $u_{\text{ip}}$ to $u$. As the layers of a neural network can be topologically ordered[1], the above path lengths are well defined and finite. Further, for any $s \in \{\text{sp,lp,rw}\}$ and $t \in \{\text{ip,op}\}$, $\delta_t^s(u)$ can be computed for all $u \in \mathcal{L}$, in $\mathcal{O}(|\mathcal{E}|)$ time (see Appendix A.3 for details).

We are now ready to describe OTMANN. Given two networks $\mathcal{G}_1 = (\mathcal{L}_1, \mathcal{E}_1), \mathcal{G}_2 = (\mathcal{L}_2, \mathcal{E}_2)$ with $n_1, n_2$ layers respectively, we will attempt to match the layer masses in both networks. We let $Z \in \mathbb{R}_+^{n_1 \times n_2}$ be such that $Z(i,j)$ denotes the amount of mass matched between layer $i \in \mathcal{G}_1$ and $j \in \mathcal{G}_2$. The OTMANN distance is computed by solving the following optimisation problem.

$$\underset{Z}{\text{minimise}} \quad \phi_{\text{lmm}}(Z) + \phi_{\text{nas}}(Z) + \nu_{\text{str}}\phi_{\text{str}}(Z) \tag{3}$$

$$\text{subject to } \sum_{j \in \mathcal{L}_2} Z_{ij} \leq \ell m(i), \ \sum_{i \in \mathcal{L}_1} Z_{ij} \leq \ell m(j), \ \forall i, j$$

The label mismatch term $\phi_{\text{lmm}}$, penalises matching masses that have different labels, while the structural term $\phi_{\text{str}}$ penalises matching masses at structurally different positions with respect to each other. If we choose not to match any mass in either network, we incur a non-assignment penalty $\phi_{\text{nas}}$. $\nu_{\text{str}} > 0$ determines the trade-off between the structural and other terms. The inequality constraints ensure that we do not over assign the masses in a layer. We now describe $\phi_{\text{lmm}}, \phi_{\text{nas}}$, and $\phi_{\text{str}}$.

*Label mismatch penalty* $\phi_{\text{lmm}}$: We begin with a label penalty matrix $M \in \mathbb{R}^{L \times L}$ where $L$ is the number of all label types and $M(\text{x}, \text{y})$ denotes the penalty for transporting a unit mass from a layer with label x to a layer with label y. We then construct a matrix $C_{\text{lmm}} \in \mathbb{R}^{n_1 \times n_2}$ with $C_{\text{lmm}}(i,j) = M(\ell\ell(i), \ell\ell(j))$ corresponding to the mislabel cost for matching unit mass from each layer $i \in \mathcal{L}_1$ to each layer $j \in \mathcal{L}_2$. We then set $\phi_{\text{lmm}}(Z) = \langle Z, C_{\text{lmm}} \rangle = \sum_{i \in \mathcal{L}_1, j \in \mathcal{L}_2} Z(i,j)C(i,j)$ to be the sum of all matchings from $\mathcal{L}_1$ to $\mathcal{L}_2$ weighted by the label penalty terms. This matrix $M$, illustrated in Table 1, is a parameter that needs to be specified for OTMANN. They can be specified with an intuitive understanding of the functionality of the layers; e.g. many values in $M$ are $\infty$, while for similar layers, we choose a value less than 1.

*Non-assignment penalty* $\phi_{\text{nas}}$: We set this to be the amount of mass that is unassigned in both networks, i.e. $\phi_{\text{nas}}(Z) = \sum_{i \in \mathcal{L}_1} \left( \ell m(i) - \sum_{j \in \mathcal{L}_2} Z_{ij} \right) + \sum_{j \in \mathcal{L}_2} \left( \ell m(j) - \sum_{i \in \mathcal{L}_1} Z_{ij} \right)$. This essentially implies that the cost for not assigning unit mass is 1. The costs in Table 1 are defined relative to this. For similar layers x, y, $M(\text{x}, \text{y}) \ll 1$ and for disparate layers $M(\text{x}, \text{

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

# A  Additional Details on OTMANN

## A.1  Optimal Transport Reformulation

We begin with a review optimal transport. Throughout this section, $\langle \cdot, \cdot \rangle$ denotes the Frobenius dot product. $\mathbf{1}_n, \mathbf{0}_n \in \mathbb{R}^n$ denote a vector of ones and zeros respectively.

**A review of Optimal Transport [47]:** Let $y_1 \in \mathbb{R}_+^{n_1}, y_2 \in \mathbb{R}_+^{n_2}$ be such that $\mathbf{1}_{n_1}^\top y_1 = \mathbf{1}_{n_2}^\top y_2$. Let $C \in \mathbb{R}_+^{n_1 \times n_2}$. The following optimisation problem,

$$\underset{Z}{\text{minimise}} \quad \langle Z, C \rangle \tag{4}$$

$$\text{subject to} \quad Z > 0, \;\; Z\mathbf{1}_{n_2} = y_1, \;\; Z^\top \mathbf{1}_{n_1} = y_2.$$

is called an *optimal transport* program. One interpretation of this set up is that $y_1$ denotes the supplies at $n_1$ warehouses, $y_2$ denotes the demands at $n_2$ retail stores, $C_{ij}$ denotes the cost of transporting a unit mass of supplies from warehouse $i$ to store $j$ and $Z_{ij}$ denotes the mass of material transported from $i$ to $j$. The program attempts to find transportation plan which minimises the total cost of transportation $\langle Z, C \rangle$.

**OT formulation of (3):** We now describe the OT formulation of the OTMANN distance. In addition to providing an efficient way to solve (3), the OT formulation will allow us to prove the metric properties of the solution. When computing the distance between $\mathcal{G}_1, \mathcal{G}_2$, for $i = 1, 2$, let $tm(\mathcal{G}_i) = \sum_{u \in \mathcal{L}_i} \ell m(u)$ denote the total mass in $\mathcal{G}_i$, and $\bar{n}_i = n_i + 1$ where $n_i = |\mathcal{L}_i|$. $y_1 = [\{\ell m(u)\}_{u \in \mathcal{L}_1}, tm(\mathcal{G}_2)] \in \mathbb{R}^{\bar{n}_1}$ will be the supplies in our OT problem, and $y_2 = [\{\ell m(u)\}_{u \in \mathcal{L}_2}, tm(\mathcal{G}_1)] \in \mathbb{R}^{\bar{n}_2}$ will be the demands. To define the cost matrix, we augment the mislabel and structural penalty matrices $C_{\text{lmm}}, C_{\text{str}}$ with an additional row and column of zeros; i.e. $C'_{\text{lmm}} = [C_{\text{lmm}} \; \mathbf{0}_{n_1}; \mathbf{0}_{\bar{n}_2}^\top 0] \in \mathbb{R}^{\bar{n}_1 \times \bar{n}_2}$; $C'_{\text{str}}$ is defined similarly. Let $C'_{\text{nas}} = [\mathbf{0}_{n_1, n_2} \; \mathbf{1}_{n_1}; \mathbf{1}_{n_2}^\top 0] \in \mathbb{R}^{\bar{n}_1 \times \bar{n}_2}$. We will show that (3) is equivalent to the following OT program.

$$\underset{Z'}{\text{minimise}} \quad \langle Z', C' \rangle \tag{5}$$

$$\text{subject to} \quad Z'\mathbf{1}_{\bar{n}_2} = y_1, \quad Z'^\top \mathbf{1}_{\bar{n}_1} = y_2.$$

One interpretation of (5) is that the last row/column appended to the cost matrices serve as a non-assignment layer and that the cost for transporting unit mass to this layer from all other layers is 1. The costs for mislabelling was defined relative to this non-assignment cost. The costs for similar layers is much smaller than 1; therefore, the optimiser is incentivised to transport mass among similar layers rather than not assign it provided that the structural penalty is not too large. Correspondingly, the cost for very disparate layers is much larger so that we would never match, say, a convolutional layer with a pooling layer. In fact, the $\infty$'s in Table 1 can be replaced by any value larger than 2 and the solution will be the same. The following theorem shows that (3) and (5) are equivalent.

**Theorem 2.** *Problems (3) and (5) are equivalent, in that they both have the same minimum and we can recover the solution of one from the other.*

*Proof.* We will show that there exists a bijection between feasible points in both problems with the same value for the objective. First let $Z \in \mathbb{R}^{n_1 \times n_2}$ be a feasible point for (3). Let $Z' \in \mathbb{R}^{\bar{n}_1 \times \bar{n}_2}$ be such that its first $n_1 \times n_2$ block is $Z$ and, $Z_{\bar{n}_1 j} = \sum_{i=1}^{n_1} Z_{ij}$, $Z_{i\bar{n}_2} = \sum_{j=1}^{n_2} Z_{ij}$, and $Z_{\bar{n}_1, \bar{n}_2} = \sum_{ij} Z_{ij}$. Then, for all $i \leq n_1$, $\sum_j Z'_{ij} = \ell m(j)$ and $\sum_j Z'_{\bar{n}_1 j} Z'_{ij} = \sum_j \ell m(j) - \sum_{ij} Z_{ij} + Z_{\bar{n}_1, \bar{n}_2} = tm(\mathcal{G}_2)$. We then have, $Z'\mathbf{1}_{\bar{n}_2} = y_1$ Similarly, we can show $Z'^\top \mathbf{1}_{\bar{n}_1} = y_2$. Therefore, $Z'$ is feasible for (5). We see that the objectives are equal via simple calculations,

$$\langle Z', C' \rangle = \langle Z', C'_{\text{lmm}} + C'_{\text{str}} \rangle + \langle Z', C'_{\text{nas}} \rangle \tag{6}$$

$$= \langle Z, C_{\text{lmm}} + C_{\text{str}} \rangle + \sum_{j=1}^{n_2} Z'_{ij} + \sum_{i=1}^{n_1} Z'_{ij}$$

$$= \langle Z, C_{\text{lmm}} \rangle + \langle Z, C_{\text{str}} \rangle + \sum_{i \in \mathcal{L}_1} \left( \ell m(i) - \sum_{j \in \mathcal{L}_2} Z_{ij} \right) + \sum_{j \in \mathcal{L}_2} \left( \ell m(j) - \sum_{i \in \mathcal{L}_1} Z_{ij} \right).$$

The converse also follows via a straightforward argument. For given $Z'$ that is feasible for (5), we let $Z$ be the first $n_1 \times n_2$ block. By the equality constraints and non-negativity of $Z'$, $Z$ is feasible for (3). By reversing the argument in (6) we see that the objectives are also equal. $\square$

Figure 3: An example of 2 CNNs which have $d = \bar{d} = 0$ distance. The OT solution matches the mass in each layer in the network on the left to the layer horizontally opposite to it on the right with 0 cost. For layer 2 on the left, its mass is mapped to layers 2 and 3 on the left. However, while the descriptor of these networks is different, their functional behaviour is the same.

## A.2 Distance Properties of OTMANN

The following theorem shows that the solution of (3) is a pseudo-distance. This is a formal version of Theorem 1 in the main text.

**Theorem 3.** *Assume that the mislabel cost matrix $M$ satisfies the triangle inequality; i.e. for all labels* x, y, z *we have* $M(\mathrm{x},\mathrm{z}) \leq M(\mathrm{x},\mathrm{y}) + M(\mathrm{y},\mathrm{z})$. *Let* $d(\mathcal{G}_1, \mathcal{G}_2)$ *be the solution of* (3) *for networks* $\mathcal{G}_1, \mathcal{G}_2$. *Then* $d(\cdot, \cdot)$ *is a pseudo-distance. That is, for all networks* $\mathcal{G}_1, \mathcal{G}_2, \mathcal{G}_3$, *it satisfies,* $d(\mathcal{G}_1, \mathcal{G}_2) > 0$, $d(\mathcal{G}_1, \mathcal{G}_2) = d(\mathcal{G}_2, \mathcal{G}_1)$, $d(\mathcal{G}_1, \mathcal{G}_1) = 0$ *and* $d(\mathcal{G}_1, \mathcal{G}_3) \leq d(\mathcal{G}_1, \mathcal{G}_2) + d(\mathcal{G}_2, \mathcal{G}_3)$.

Some remarks are in order. First, observe that while $d(\cdot, \cdot)$ is a pseudo-distance, it is not a distance; i.e. $d(\mathcal{G}_1, \mathcal{G}_2) = 0 \not\Rightarrow \mathcal{G}_1 = \mathcal{G}_2$. For example, while the networks in Figure 3 have different descriptors according to our formalism in Section 2.2, their distance is 0. However, it is not hard to see that their functionality is the same – in both cases, the output of layer 1 is passed through 16 `conv3` filters and then fed to a layer with 32 `conv3` filters – and hence, this property is desirable in this example. It is not yet clear however, if the topology induced by our metric equates two functionally dissimilar networks. We leave it to future work to study equivalence classes induced by the OTMANN distance. Second, despite the OT formulation, this is not a Wasserstein distance. In particular, the supports of the masses and the cost matrices change depending on the two networks being compared.

***Proof of Theorem 3.*** We will use the OT formulation (5) in this proof. The first three properties are straightforward. Non-negativity follows from non-negativity of $Z', C'$ in (5). It is symmetric since the cost matrix for $d(\mathcal{G}_2, \mathcal{G}_1)$ is $C'^\top$ if the cost matrix for $d(\mathcal{G}_1, \mathcal{G}_2)$ is $C$ and $\langle Z', C' \rangle = \langle Z'^\top, C'^\top \rangle$ for all $Z'$. We also have $d(\mathcal{G}_1, \mathcal{G}_1) = 0$ since, then, $C'$ has a zero diagonal.

To prove the triangle inequality, we will use a gluing lemma, similar to what is used in the proof of Wasserstein distances [33]. Let $\mathcal{G}_1, \mathcal{G}_2, \mathcal{G}_3$ be given and $m_1, m_2, m_3$ be their total masses. Let the solutions to $d(\mathcal{G}_1, \mathcal{G}_2)$ and $d(\mathcal{G}_2, \mathcal{G}_3)$ be $P \in \mathbb{R}^{\bar{n}_1 \times \bar{n}_2}$ and $Q \in \mathbb{R}^{\bar{n}_2 \times \bar{n}_3}$ respectively. When solving (5), we see that adding extra mass to the non-assignment layers does not change the objective, as an optimiser can transport mass between the two layers with 0 cost. Hence, we can assume w.l.o.g that (5) was solved with $y_i = \left[ \{\ell m(u)\}_{u \in \mathcal{L}_i}, \left( \sum_{j \in \{1,2,3\}} tm(\mathcal{G}_j) - tm(\mathcal{G}_i) \right) \right] \in \mathbb{R}^{\bar{n}_i}$ for $i = 1, 2, 3$, when computing the distances $d(\mathcal{G}_1, \mathcal{G}_2), d(\mathcal{G}_1, \mathcal{G}_3), d(\mathcal{G}_2, \mathcal{G}_3)$; i.e. the total mass was $m_1 + m_2 + m_3$ for all three pairs. We can similarly assume that $P, Q$ account for this extra mass, i.e. $P_{\bar{n}_1 \bar{n}_2}$ and $Q_{\bar{n}_2 \bar{n}_3}$ have been increased by $m_3$ and $m_1$ respectively from their solutions in (5).

To apply the gluing lemma, let $S = P\mathrm{diag}(1/y_2)Q \in \mathbb{R}^{\bar{n}_1 \times \bar{n}_3}$, where $\mathrm{diag}(1/y_2)$ is a diagonal matrix whose $(j, j)^{\mathrm{th}}$ element is $1/(y_2)_j$ (note $y_2 > 0$). We see that $S$ is feasible for (5) when computing $d(\mathcal{G}_1, \mathcal{G}_3)$,

$$R\mathbf{1}_{\bar{n}_3} = P\mathrm{diag}(1/y_2)Q\mathbf{1}_{\bar{n}_3} = P\mathrm{diag}(1/y_2)y_2 = P\mathbf{1}_{\bar{n}_2} = y_1.$$

Similarly, $R^\top \mathbf{1}_{\bar{n}_1} = y_3$. Now, let $U', V', W'$ be the cost matrices $C'$ in (5) when computing $d(\mathcal{G}_1, \mathcal{G}_2), d(\mathcal{G}_2, \mathcal{G}_3)$, and $d(\mathcal{G}_1, \mathcal{G}_3)$ respectively. We will use the following technical lemma whose proof is given below.

**Lemma 4.** *For all* $i \in \mathcal{L}_1$, $j \in \mathcal{L}_2$, $k \in \mathcal{L}_3$, *we have* $W'_{ik} \leq U'_{ij} + V'_{jk}$.

Applying Lemma 4 yields the triangle inequality.

$$d(\mathcal{G}_1, \mathcal{G}_3) \leq \langle R, W' \rangle = \sum_{i \in \mathcal{L}_1, k \in \mathcal{L}_3} W'_{ik} \sum_{j \in \mathcal{L}_2} \frac{P_{ij} Q_{jk}}{(y_2)_j} \leq \sum_{i,j,k} (U'_{ij} + V'_{jk}) \frac{P_{ij} Q_{jk}}{(y_2)_j}$$

$$= \sum_{ij} \frac{U'_{ij} P_{ij}}{(y_2)_j} \sum_k Q_{jk} + \sum_{jk} \frac{V'_{jk} Q_{jk}}{(y_2)_j} \sum_k P_{ij}$$

$$= \sum_{ij} U'_{ij} P_{ij} + \sum_{jk} V'_{jk} Q_{jk} = d(\mathcal{G}_1, \mathcal{G}_2) + d(\mathcal{G}_2, \mathcal{G}_3)$$

The first step uses the fact that $d(\mathcal{G}_1, \mathcal{G}_3)$ is the minimum of all feasible solutions and the third step uses Lemma 4. The fourth step rearranges terms and the fifth step uses $P^\top \mathbf{1}_{\bar{n}_1} = Q \mathbf{1}_{\bar{n}_3} = y_2$.     $\square$

***Proof of Lemma 4.*** Let $W' = W'_{\text{lmm}} + W'_{\text{str}} + W'_{\text{nas}}$ be the decomposition into the label mismatch, structural and non-assignment parts of the cost matrices; define similar quantities $U'_{\text{lmm}}, U'_{\text{str}}, U'_{\text{nas}}, V'_{\text{lmm}}, V'_{\text{str}}, V'_{\text{nas}}$ for $U', V'$. Noting $a \leq b+c$ and $d \leq e+f$ implies $a+d \leq b+e+c+f$, it is sufficient to show the triangle inequality for each component individually. For the label mismatch term, $(W'_{\text{lmm}})_{ik} \leq (U'_{\text{lmm}})_{ij} + (V'_{\text{lmm}})_{jk}$ follows directly from the conditions on $M$ by setting $\mathtt{x} = \ell\ell(i), \mathtt{y} = \ell\ell(j), \mathtt{z} = \ell\ell(k)$, where $i, j, k$ are indexing in $\mathcal{L}_1, \mathcal{L}_2, \mathcal{L}_3$ respectively.

For the non-assignment terms, when $(W'_{\text{nas}})_{ik} = 0$ the claim is true trivially. $(W'_{\text{nas}})_{ik} = 1$, either when $(i = \bar{n}_1, k \leq n_3)$ or $(i \leq n_1, k = \bar{n}_3)$. In the former case, when $j \leq n_2$, $(U'_{\text{nas}})_{jk} = 1$ and when $j = \bar{n}_2$, $(V'_{\text{nas}})_{\bar{n}_2} = 1$ as $k \leq n_3$. We therefore have, $(W'_{\text{nas}})_{ik} = (U'_{\text{nas}})_{ij} + (V'_{\text{nas}})_{jk} = 1$. A similar argument shows equality for the $(i \leq n_1, k = \bar{n}_3)$ case as well.

Finally, for the structural terms we note that $W'_{\text{str}}$ can be written as $W'_{\text{str}} = \sum_t W'^{(t)}$ as can $U'^{(t)}, T'^{(t)}$. Here $t$ indexes over the choices for the types of distances considered, i.e. $t \in \{\text{sp, lp, rw}\} \times \{\text{ip, op}\}$. It is sufficient to show $(W'^{(t)})_{ik} \leq (U'^{(t)})_{ij} + (T'^{(t)})_{jk}$. This inequality takes the form,

$$|\delta_{1i}^{(t)} - \delta_{3k}^{(t)}| \leq |\delta_{1i}^{(t)} - \delta_{2j}^{(t)}| + |\delta_{2j}^{(t)} - \delta_{3k}^{(t)}|.$$

Where $\delta_{g\ell}^{(t)}$ refers to distance type $t$ in network $g$ for layer $s$. The above is simply the triangle inequality for real numbers. This concludes the proof of Lemma 4.     $\square$

## A.3   Implementation & Design Choices

**Masses on the decision & input/output layers:** It is natural to ask why one needs to model the mass in the decision and input/output layers. For example, a seemingly natural choice is to use $0$ for these layers. Using $0$ mass, is a reasonable strategy if we were to allow only one decision layer. However, when there are multiple decision layers, consider comparing the following two networks: the first has a feed forward MLP with non-linear layers, the second is the same network but with an additional linear decision layer $u$, with one edge from $u_{\text{ip}}$ to $u$ and an edge from $u$ to $u_{\text{op}}$. This latter models the function as a linear + non-linear term which might be suitable for some problems unlike modeling it only as a non-linear term. If we do not add layer masses for the input/output/decision layers, then the distance between both networks would be $0$ - as there will be equal mass in the FF part for both networks and they can be matched with $0$ cost.

---

**Algorithm 1:** Compute $\delta_{\text{op}}^{\text{rw}}(u)$ for all $u \in \mathcal{L}$

---

**Require:** $\mathcal{G} = (\mathcal{L}, \mathcal{E})$, $\mathcal{L}$ is topologically sorted in $S$.
1: $\delta_{\text{op}}^{\text{rw}}(u_{\text{op}}) = 0$, $\delta_{\text{op}}^{\text{rw}}(u) = \mathtt{nan} \ \forall u \neq u_{\text{op}}$.
2: **while** $S$ is not empty **do**
3:     $u \leftarrow \text{pop\_last}(S)$
4:     $\Delta \leftarrow \{\delta_{\text{op}}^{\text{rw}}(c) : c \in \text{children}(u)\}$
5:     $\delta_{\text{op}}^{\text{rw}}(u) \leftarrow 1 + \text{average}(\Delta)$
6: **end while**
7: **Return** $\delta_{\text{op}}^{\text{rw}}$.

---

| | c3 | c5 | c7 | mp | ap | fc | sm |
|---|---|---|---|---|---|---|---|
| c3 | 0 | 0.2 | 0.3 | | | | |
| c5 | 0.2 | 0 | 0.2 | | | | |
| c7 | 0.3 | 0.2 | 0 | | | | |
| mp | | | | 0 | 0.25 | | |
| ap | | | | 0.25 | 0 | | |
| fc | | | | | | 0 | |
| sm | | | | | | | 0 |

Table 4: The label mismatch cost matrix $M$ we used in our CNN experiments. $M(\mathtt{x}, \mathtt{y})$ denotes the penalty for transporting a unit mass from a layer with label $\mathtt{x}$ to a layer with label $\mathtt{y}$. The labels abbreviated are `conv3`, `conv5`, `conv7`, `max-pool`, `avg-pool`, `fc`, and `softmax` in order. A blank indicates $\infty$ cost. We have not shown the `ip` and `op` layers, but they are similar to the `fc` column, 0 in the diagonal and $\infty$ elsewhere.

| | re | cr | \<rec\> | lg | ta | lin |
|---|---|---|---|---|---|---|
| re | 0 | .1 | .1 | .25 | .25 | |
| cr | .1 | 0 | .1 | .25 | .25 | |
| \<rec\> | .1 | .1 | 0 | .25 | .25 | |
| lg | .25 | .25 | .25 | 0 | .1 | |
| ta | .25 | .25 | .25 | .1 | 0 | |
| lin | | | | | | 0 |

Table 5: The label mismatch cost matrix $M$ we used in our MLP experiments. The labels abbreviated are `relu`, `crelu`, `<rec>`, `logistic`, `tanh`, and `linear` in order. `<rec>` is place-holder for any other rectifier such as `leaky-relu`, `softplus`, `elu`. A blank indicates $\infty$ cost. The design here was simple. Each label gets 0 cost with itself. A rectifier gets 0.1 cost with another rectifier and 0.25 with a sigmoid; vice versa for all sigmoids. The rest of the costs are infinity. We have not shown the `ip` and `op`, but they are similar to the `lin` column, 0 in the diagonal and $\infty$ elsewhere.

**Computing path lengths $\delta_s^t$:** Algorithm 1 computes all path lengths in $O(|\mathcal{E}|)$ time. Note that topological sort of a connected digraph also takes $O(|\mathcal{E}|)$ time. The topological sorting ensures that $\delta_{\mathrm{op}}^{\mathrm{rw}}$ is always computed for the children in step 4. For $\delta_{\mathrm{op}}^{\mathrm{sp}}, \delta_{\mathrm{op}}^{\mathrm{lp}}$ we would replace the averaging of $\Delta$ in step 5 with the minimum and maximum of $\Delta$ respectively.

For $\delta_{\mathrm{ip}}^{\mathrm{rw}}$ we make the following changes to Algorithm 1. In step 1, we set $\delta_{\mathrm{ip}}^{\mathrm{rw}}(u_{\mathrm{ip}}) = 0$, in step 3, we pop_first and $\Delta$ in step 4 is computed using the parents. $\delta_{\mathrm{ip}}^{\mathrm{sp}}, \delta_{\mathrm{ip}}^{\mathrm{lp}}$ are computed with the same procedure but by replacing the averaging with minimum or maximum as above.

**Label Penalty Matrices:** The label penalty matrices used in our NASBOT implementation, described below, satisfy the triangle inequality condition in Theorem 3.

CNNs: Table 4 shows the label penalty matrix $M$ for used in our CNN experiments with labels `conv3`, `conv5`, `conv7`, `max-pool`, `avg-pool`, `softmax`, `ip`, `op`. `conv`$k$ denotes a $k \times k$ convolution while `avg-pool` and `max-pool` are pooling operations. In addition, we also use `res3`, `res5`, `res7` layers which are inspired by ResNets. A `res`$k$ uses 2 concatenated `conv`$k$ layers but the input to the first layer is added to the output of the second layer before the relu activation – See Figure 2 in He et al. [12]. The layer mass for `res`$k$ layers is twice that of a `conv`$k$ layer. The costs for the `res` in the label penalty matrix is the same as the `conv` block. The cost between a `res`$k$ and `conv`$j$ is $M(\mathtt{res}k, \mathtt{conv}j) = 0.9 \times M(\mathtt{conv}k, \mathtt{conv}j) + 0.1 \times 1$; i.e. we are using a convex combination of the `conv` costs and the non-assignment cost. The intuition is that a `res`$k$ is similar to `conv`$k$ block except for the residual addition.

MLPs: Table 5 shows the label penalty matrix $M$ for used in our MLP experiments with labels `relu`, `crelu`, `leaky-relu`, `softplus`, `elu`, `logistic`, `tanh`, `linear`, `ip`, `op`. Here the first seven are common non-linear activations; `relu`, `crelu`, `leaky-relu`, `softplus`, `elu` rectifiers while `logistic` and `tanh` are sigmoidal activations.

**Other details:** Our implementation of OTMANN differs from what is described in the main text in two ways. First, in our CNN experiments, for a `fc` layer $u$, we use $0.1 \times \ell m(u) \times \langle \text{\#-incoming-channels} \rangle$ as the mass, i.e. we multiply it by $0.1$ from what is described in the main text. This is because, in the convolutional and pooling channels, each unit is an image where as in the `fc` layers each unit is a scalar. One could, in principle, account for the image sizes at the various layers when computing the layer masses, but this also has the added complication of depending on the size of the input image which varies from problem to problem. Our approach is simpler and yields reasonable results.

Secondly, we use a slightly different form for $C_{\text{str}}$. First, for $i \in \mathcal{L}_1$, $j \in \mathcal{L}_2$, we let $C_{\text{str}}^{\text{all}}(i,j) = \frac{1}{6} \sum_{s \in \{\text{sp, lp, rw}\}} \sum_{t \in \{\text{ip,op}\}} |\delta_t^s(i) - \delta_t^s(j)|$ be the average of *all* path length differences; i.e. $C_{\text{str}}^{\text{all}}$ captures the path length differences when considering all layers. For CNNs, we similarly construct matrices $C_{\text{str}}^{\text{conv}}, C_{\text{str}}^{\text{pool}}, C_{\text{str}}^{\text{fc}}$, except they only consider the convolutional, pooling and fully connected layers respectively in the path lengths. For $C_{\text{str}}^{\text{conv}}$, the distances to the output (from the input) can be computed by zeroing outgoing (incoming) edges to layers that are not convolutional. We can similarly construct $C_{\text{str}}^{\text{pool}}$ and $C_{\text{str}}^{\text{fc}}$ only counting the pooling and fully connected layers. Our final cost matrix for the structural penalty is the average of these four matrices, $C_{\text{str}} = (C_{\text{str}}^{\text{all}} + C_{\text{str}}^{\text{conv}} + C_{\text{str}}^{\text{pool}} + C_{\text{str}}^{\text{fc}})/4$. For MLPs, we adopt a similar strategy by computing matrices $C_{\text{str}}^{\text{all}}, C_{\text{str}}^{\text{rec}}, C_{\text{str}}^{\text{sig}}$ with all layers, only rectifiers, and only sigmoidal layers and let $C_{\text{str}} = (C_{\text{str}}^{\text{all}} + C_{\text{str}}^{\text{rec}} + C_{\text{str}}^{\text{sig}})/3$. The intuition is that by considering certain types of layers, we are accounting for different types of information flow due to different operations.

### A.4    Some Illustrations of the OTMANN Distance

We illustrate that OTMANN computes reasonable distances on neural network architectures via a two-dimensional t-SNE visualisation [27] of the network architectures based. Given a distance matrix between $m$ objects, t-SNE embeds them in a $d$ dimensional space so that objects with small distances are placed closer to those that have larger distances. Figure 4 shows the t-SNE embedding using the OTMANN distance and its noramlised version. We have indexed 13 networks in both figures in a-n and displayed their architectures in Figure 5. Similar networks are placed close to each other indicating that OTMANN induces a meaningful topology among neural network architectures.

Next, we show that the distances induced by OTMANN are correlated with validation error performance. In Figure 6 we provide the following scatter plot for networks trained in our experiments for the Indoor, Naval and Slice datasets. Each point in the figure is for pair of networks. The $x$-axis is the OTMANN distance between the pair and the $y$-axis is the difference in the validation error on the dataset. In each figure we used $300$ networks giving rise to $45K$ pairwise points in each scatter plot. As the figure indicates, when the distance is small the difference in performance is close to $0$. However, as the distance increases, the points are more scattered. Intuitively, one should expect that while networks that are far apart could perform similarly or differently, similar networks should perform similarly. Hence, OTMANN induces a useful topology in the space of architectures that is smooth for validaiton performance on real world datasets. This demonstrates that it can be incorporated in a BO framework to optimise a network based on its validation error.

## B    Implementation of NASBOT

Here, we describe our BO framework for NASBOT in full detail.

### B.1    The Kernel

As described in the main text, we use a negative exponentiated distance as our kernel. Precisely, we use,

$$\kappa(\cdot, \cdot) = \alpha e^{-\sum_i \beta_i d_i^p(\cdot, \cdot)} + \bar{\alpha} e^{-\sum_i \bar{\beta}_i \bar{d}_i^{\bar{p}}(\cdot, \cdot)}. \tag{7}$$

Here, $d_i, \bar{d}_i$, are the OTMANN distance and its normalised counterpart developed in Section 3, computed with different values for $\nu_{\text{str}} \in \{\nu_{\text{str},i}\}_i$. $\beta_i, \bar{\beta}_i$ manage the relative contributions of $d_i, \bar{d}_i$, while $(\alpha, \bar{\alpha})$ manage the contributions of each kernel in the sum. An ensemble approach of the above form, instead of trying to pick a single best value, ensures that NASBOT accounts for the different topologies induced by the different distances $d_i, \bar{d}_i$. In the experiments we report, we used $\{\nu_{\text{str},i}\}_i = \{0.1, 0.2, 0.4, 0.8\}$, $p = 1$ and $\bar{p} = 2$. Our experience suggests that NASBOT was not particularly sensitive to these choices expect when we used only very large or only very small values in $\{\nu_{\text{str},i}\}_i$.

NASBOT, as described above has 11 hyper-parameters of its own; $\alpha, \bar{\alpha}, \{(\beta_i, \bar{\beta}_i)\}_{i=1}^4$ and the GP noise variance $\eta^2$. While maximising the GP marginal likelihood is a common approach to pick hyper-parameters, this might cause over-fitting when there are many of them. Further, as training large neural networks is typically expensive, we have to content with few observations for the GP

Figure 4: Two dimensional t-SNE embeddings of 100 randomly generated CNN architectures based on the OTMANN distance (top) and its normalised version (bottom). Some networks have been indexed a-n in the figures; these network architectures are illustrated in Figure 5. Networks that are similar are embedded close to each other indicating that the OTMANN induces a meaningful topology among neural network architectures.

Figure 5: Illustrations of the nextworks indexed a-n in Figure 4.

Figure 6: Each point in the scatter plot indicates the log distance between two architectures ($x$ axis) and the difference in the validation error ($y$ axis), on the Indoor, Naval and Slice datasets. We used 300 networks, giving rise to $\sim 45K$ pairwise points. On all datasets, when the distance is small, so is the difference in the validation error. As the distance increases, there is more variance in the validation error difference. Intuitively, one should expect that while networks that are far apart could perform similarly or differently, networks with small distance should perform similarly.

in practical settings. Our solution is to start with a (uniform) prior over these hyper-parameters and sample hyper-parameter values from the posterior under the GP likelihood [40], which we found to be robust. While it is possible to treat $\nu_{\text{str}}$ itself as a hyper-parameter of the kernel, this will require us to re-compute all pairwise distances of networks that have already been evaluated each time we change the hyper-parameters. On the other hand, with the above approach, we can compute and store distances for different $\nu_{\text{str},i}$ values whenever a new network is evaluated, and then compute the kernel cheaply for different values of $\alpha, \bar{\alpha}, \{(\beta_i, \bar{\beta}_i)\}_i$.

## B.2  Optimising the Acquisition

We use a evolutionary algorithm (EA) approach to optimise the acquisition function (2). We begin with an initial pool of networks and evaluate the acquisition $\varphi_t$ on those networks. Then we generate a set of $N_{\text{mut}}$ mutations of this pool as follows. First, we stochastically select $N_{\text{mut}}$ candidates from the set of networks already evaluated such that those with higher $\varphi_t$ values are more likely to be selected than those with lower values. Then we apply a mutation operator to each candidate, to produce a modified architecture. Finally, we evaluate the acquisition on this $N_{\text{mut}}$ mutations, add it to the initial pool, and repeat for the prescribed number of steps.

**Mutation Operator:** To describe the mutation operator, we will first define a library of modifications to a neural network. These modifications, described in Table 6, might change the architecture either by increasing or decreasing the number of computational units in a layer, by adding or deleting layers, or by changing the connectivity of existing layers. They provide a simple mechanism to explore the space of architectures that are close to a given architecture. The *one-step mutation operator* takes a given network and applies one of the modifications in Table 6 picked at random to produce a new network. The *k-step mutation operator* takes a given network, and repeatedly applies the one-step operator $k$ times – the new network will have undergone $k$ changes from the original one. One can also define a compound operator, which picks the number of steps probabilistically. In our implementation of NASBOT, we used such a compound operator with probabilities $(0.5, 0.25, 0.125, 0.075, 0.05)$; i.e. it chooses a one-step operator with probability $0.5$, a 4-step operator with probability $0.075$, etc. Typical implementations of EA in Euclidean spaces define the mutation operator via a Gaussian (or other) perturbation of a chosen candidate. It is instructive to think of the probabilities for each step in our scheme above as being analogous to the width of the Gaussian chosen for perturbation.

**Sampling strategy:** The sampling strategy for EA is as follows. Let $\{z_i\}_i$, where $z_i \in \mathcal{X}$ be the points evaluated so far. We sample $N_{\text{mut}}$ new points from a distribution $\pi$ where $\pi(z_i) \propto \exp(g(z_i)/\sigma)$. Here $g$ is the function to be optimised (for NASBOT, $\varphi_t$ at time $t$). $\sigma$ is the standard deviation of all previous evaluations. As the probability for large $g$ values is higher, they are more likely to get selected. $\sigma$ provides normalisation to account for different ranges of function values.

| Operation | Description |
|---|---|
| dec_single | Pick a layer at random and decrease the number of units by $1/8$. |
| dec_en_masse | First topologically order the networks, randomly pick $1/8$ of the layers (in order) and decrease the number of units by $1/8$. For networks with eight layers or fewer pick a $1/4$ of the layers (instead of $1/8$) and for those with four layers or fewer pick $1/2$. |
| inc_single | Pick a layer at random and increase the number of units by $1/8$. |
| inc_en_masse | Choose a large sub set of layers, as for dec_en_masse, and increase the number of units by $1/8$. |
| dup_path | This modifier duplicates a random path in the network. Randomly pick a node $u_1$ and then pick one of its children $u_2$ randomly. Keep repeating to generate a path $u_1, u_2, \ldots, u_{k-1}, u_k$ until you decide to stop randomly. Create duplicate layers $\tilde{u}_2, \ldots, \tilde{u}_{k-1}$ where $\tilde{u}_i = u_i$ for $i = 2, \ldots, k-1$. Add these layers along with new edges $(u_1, \tilde{u}_2)$, $(\tilde{u}_{k-1}, u_k)$, and $(\tilde{u}_j, \tilde{u}_{j+1})$ for $j = 2, \ldots, k-2$. |
| remove_layer | Picks a layer at random and removes it. If this layer was the only child (parent) of any of its parents (children) $u$, then adds an edge from $u$ (one of its parents) to one of its children ($u$). |
| skip | Randomly picks layers $u, v$ where $u$ is topologically before $v$ and $(u, v) \notin \mathcal{E}$. Add $(u, v)$ to $\mathcal{E}$. |
| swap_label | Randomly pick a layer and change its label. |
| wedge_layer | Randomly pick any edge $(u, v) \in \mathcal{E}$. Create a new layer $w$ with a random label $\ell\ell(w)$. Remove $(u, v)$ from $\mathcal{E}$ and add $(u, w), (w, v)$. If applicable, set the number of units $\ell u(w)$ to be $(\ell u(u) + \ell u(v))/2$. |

Table 6: Descriptions of modifiers to transform one network to another. The first four change the number of units in the layers but do not change the architecture, while the last five change the architecture.

Since our candidate selection scheme at each step favours networks that have high acquisition value, our EA scheme is more likely to search at regions that are known to have high acquisition. The stochasticity in this selection scheme and the fact that we could take multiple steps in the mutation operation ensures that we still sufficiently explore the space. Since an evaluation of $\varphi_t$ is cheap, we can use many EA steps to explore several architectures and optimise $\varphi_t$.

**Other details:** The EA procedure is also initialised with the same initial pools in Figures 20, 21. In our NASBOT implementation, we increase the total number of EA evaluations $n_{\mathsf{EA}}$ at rate $\mathcal{O}(\sqrt{t})$ where $t$ is the current time step in NASBOT. We set $N_{\mathrm{mut}}$ to be $\mathcal{O}(\sqrt{n_{\mathsf{EA}}})$. Hence, initially we are only considering a small neighborhood around the initial pool, but as we proceed along BO, we expand to a larger region, and also spend more effort to optimise $\varphi_t$.

**Considerations when performing modifications:** The modifications in Table 6 is straightforward in MLPs. But in CNNs one needs to ensure that the image sizes are the same when concatenating them as an input to a layer. This is because strides can shrink the size of the image. When we perform a modification we check if this condition is violated and if so, disallow that modification. When a skip modifier attempts to add a connection from a layer with a large image size to one with a smaller one, we add `avg-pool` layers at stride 2 so that the connection can be made (this can be seen, for e.g. in the second network in Fig. 8).

### B.3   Other Implementation Details

**Initialisation:** We initialise NASBOT (and other methods) with an initial pool of 10 networks. These networks are illustrated in Fig. 20 for CNNs and Fig. 21 for MLPs at the end of the document. These are the same networks used to initialise the EA procedure to optimise the acquisition. All initial networks have feed forward structure. For the CNNs, the first 3 networks have structure similar to the VGG nets [37] and the remaining have blocked feed forward structures as in He et al. [12]. We also use blocked structures for the MLPs with the layer labels decided arbitrarily.

**Domain:** For NASBOT, and other methods, we impose the following constraints on the search space. If the EA modifier (explained below) generates a network that violates these constraints, we simply skip it.

- Maximum number of layers: 60

- Maximum mass: $10^8$
- Maximum in/out degree: $5$
- Maximum number of edges: $200$
- Maximum number of units per layer: $1024$
- Minimum number of units per layer: $8$

**Layer types:** We use the layer types detailed in Appendix A.3 for both CNNs and MLPs. For CNNs, all pooling operations are done at stride 2. For convolutional layers, we use either stride 1 or 2 (specified in the illustrations). For all layers in a CNN we use `relu` activations.

**Parallel BO:** We use a parallelised experimental set up where multiple models can be evaluated in parallel. We handle parallel BO via the hallucination technique in Ginsbourger et al. [10].

Finally, we emphasise that many of the above choices were made arbitrarily, and we were able to get NASBOT working efficiently with our first choice for these parameters/specifications. Note that many end-to-end systems require specification of such choices.

## C    Addendum to Experiments

### C.1    Baselines

RAND: Our RAND implementation, operates in exactly the same way as NASBOT, except that the EA procedure (Sec. B.2) is fed a random sample from $\mathrm{Unif}(0, 1)$ instead of the GP acquisition each time it evaluates an architecture. That is, we follow the same schedule for $n_{\mathsf{EA}}$ and $N_{\mathrm{mut}}$ as we did for NASBOT . Hence RAND has the opportunity to explore the same space as NASBOT, but picks the next evaluation randomly from this space.

EA: This is as described in Appendix B except that we fix $N_{\mathrm{mut}} = 10$ all the time. In our experiments where we used a budget based on time, it was difficult to predict the total number of evaluations so as to set $N_{\mathrm{mut}}$ in perhaps a more intelligent way.

TreeBO: As the implementation from Jenatton et al. [15] was not made available, we wrote our own. It differs from the version described in the paper in a few ways. We do not tune for a regularisation penalty and step size as they do to keep it line with the rest of our experimental set up. We set the depth of the network to 60 as we allowed 60 layers for the other methods. We also check for the other constraints given in Appendix B before evaluating a network. The original paper uses a tree structured kernel which can allow for efficient inference with a large number of samples. For simplicity, we construct the entire kernel matrix and perform standard GP inference. The result of the inference is the same, and the number of GP samples was always below 120 in our experiments so a sophisticated procedure was not necessary.

### C.2    Details on Training

In all methods, for each proposed network architecture, we trained the network on the train data set, and periodically evaluated its performance on the validation data set. For MLP experiments, we optimised network parameters using stochastic gradient descent with a fixed step size of $10^{-5}$ and a batch size of 256 for 20,000 iterations. We computed the validation set MSE every 100 iterations; from this we returned the minimum MSE that was achieved. For CNN experiments, we optimised network parameters using stochastic gradient descent with a batch size of 32. We started with a learning rate of $0.01$ and reduced it gradually. We also used batch normalisation and trained the model for $60,000$ batch iterations. We computed the validation set classification error every $4000$ iterations; from this we returned the minimum classification error that was achieved.

After each method returned an optimal neural network architecture, we again trained each optimal network architecture on the train data set, periodically evaluated its performance on the validation data set, and finally computed the MSE or classification error on the test data set. For MLP experiments, we used the same optimisation procedure as above; we then computed the test set MSE at the iteration where the network achieved the minimum validation set MSE. For CNN experiments, we used the same optimisation procedure as above, except here the optimal network architecture was trained

for 120,000 iterations; we then computed the test set classification error at the iteration where the network achieved the minimum validation set classification error.

### C.3    Optimal Network Architectures and Initial Pool

Here we illustrate and compare the optimal neural network architectures found by different methods. In Figures 8-11, we show some optimal network architectures found on the Cifar10 data by NASBOT, EA, RAND, and TreeBO, respectively. We also show some optimal network architectures found for these four methods on the Indoor data, in Figures 12-15, and on the Slice data, in Figures 16-19. A common feature among all optimal architectures found by NASBOT was the presence of long skip connections and multiple decision layers.

In Figure 21, we show the initial pool of MLP network architectures, and in Figure 20, we show the initial pool of CNN network architectures. On the Cifar10 dataset, VGG-19 was one of the networks in the initial pool. While all methods beat VGG-19 when trained for 24K iterations (the number of iterations we used when picking the model), TreeBO and RAND lose to VGG-19 (see Section 5 for details). This could be because the performance after shorter training periods may not exactly correlate with performance after longer training periods.

### C.4    Ablation Studies and Design Choices

We conduct experiments comparing the various design choices in NASBOT. Due to computational constraints, we carry them out on synthetic functions.

In Figure 7a, we compare NASBOT using only the normalised distance, only the unnormalised distance, and the combined kernel as in (7). While the individual distances performs well, the combined form outperforms both.

Next, we modify our EA procedure to optimise the acquisition. We execute NASBOT using only the EA modifiers which change the computational units (first four modifiers in Table 6), then using the modifiers which only change the structure of the networks (bottom 5 in Table 6), and finally using all 9 modifiers, as used in all our experiments. The combined version outperforms the first two.

Finally, we experiment with different choices for $p$ and $\bar{p}$ in (7). As the figures indicate, the performance was not particularly sensitive to these choices.

Below we describe the three synthetic functions $f_1, f_2, f_3$ used in our synthetic experiments. $f_3$ applies for CNNs while $f_1, f_2$ apply for MLPs. Here $am$ denotes the average mass per layer, $\deg_i$ is the average in degree the layers, $\deg_o$ is the average out degree, $\delta$ is the shortest distance from $u_{ip}$ to $u_{op}$, $str$ is the average stride in CNNS, $frac\_conv3$ is the fraction of layers that are `conv3`, $frac\_sigmoid$ is the fraction of layers that are sigmoidal.

$$
\begin{aligned}
f_0 &= \exp(-0.001 * |am - 1000|) + \exp(-0.5 * |\deg_i - 5|) + \exp(-0.5 * |\deg_o - 5|) + \\
&\quad \exp(-0.1 * |\delta - 5|) + \exp(-0.1 * ||\mathcal{L}| - 30|) + \exp(-0.05 * ||\mathcal{E}| - 100|) \\
f_1 &= f_0 + \exp(-3 * |str - 1.5|) + \exp(-0.3 * ||\mathcal{L}| - 50|) + \\
&\quad \exp(-0.001 * |am - 500|) + frac\_conv3 \\
f_2 &= f_0 + \exp(-0.001 * |am - 2000|) + \exp(-0.1 * ||\mathcal{E}| - 50|) + frac\_sigmoid \\
f_3 &= f_0 + frac\_sigmoid
\end{aligned}
$$

## D    Additional Discussion on Related Work

Historically, evolutionary (genetic) algorithms (EA) have been the most common method used for designing architectures [8, 21, 26, 30, 36, 41, 51]. EA techniques are popular as they provide a simple mechanism to explore the space of architectures by making a sequence of changes to networks that have already been evaluated. However, as we will discuss later, EA algorithms, while conceptually and computationally simple, are typically not best suited for optimising functions that are expensive to evaluate. A related line of work first sets up a search space for architectures via incremental modifications, and then explores this space via random exploration, MCTS, or A* search [6, 25, 32].

Figure 7: We compare NASBOT for different design choices in our framework. (a): Comparison of NASBOT using only the normalised distance $e^{-\beta\bar{d}}$, only the unnormalised distance $d^{-\beta d}$, and the combination $e^{-\beta d}+e^{-\bar{\beta}\bar{d}}$. (b): Comparison of NASBOT using only the EA modifiers which change the computational units (top 4 in Table 6), modifiers which only change the structure of the networks (bottom 5 in Table 6), and all 9 modifiers. (c): Comparison of NASBOT with different choices for $p$ and $\bar{p}$. In all figures, the $x$ axis is the number of evaluations and the $y$ axis is the negative maximum value (lower is better). All figures were produced by averaging over at least 10 runs.

Some of the methods above can only optimise among feed forward structures, e.g. Fig. 1a, but cannot handle spaces with arbitrarily structured networks, e.g. Figs. 1b, 1c.

The most successful recent architecture search methods that can handle arbitrary structures have adopted reinforcement learning (RL) [1, 52–54]. However, architecture search is in essence an *optimisation* problem – find the network with the highest function value. There is no explicit need to maintain a notion of state and solve the credit assignment problem in RL [43]. Since RL is fundamentally more difficult than optimisation [16], these methods typically need to try a very large number of architectures to find the optimum. This is not desirable, especially in computationally constrained settings.

Figure 8: Optimal network architectures found with NASBOT on Cifar10 data.

Figure 9: Optimal network architectures found with EA on Cifar10 data.

Figure 10: Optimal network architectures found with RAND on Cifar10 data.

Figure 11: Optimal network architectures found with TreeBO on Cifar10 data.

Figure 12: Optimal network architectures found with NASBOT on Indoor data.

Figure 13: Optimal network architectures found with EA on Indoor data.

Figure 14: Optimal network architectures found with RAND on Indoor data.

Figure 15: Optimal network architectures found with TreeBO on Indoor data.

Figure 16: Optimal network architectures found with NASBOT on Slice data.

Figure 17: Optimal network architectures found with EA on Slice data.

Figure 18: Optimal network architectures found with RAND on Slice data.

Figure 19: Optimal network architectures found with TreeBO on Slice data.

Figure 20: Initial pool of CNN network architectures. The first 3 networks have structure similar to the VGG nets [37] and the remaining have blocked feed forward structures as in He et al. [12].

Figure 21: Initial pool of MLP network architectures.