[Reviews · NeurIPS 2018]

Reviewer 1



This paper proposed a bayesian optimization framework for Neural Architecture Search (NAS). The key contribution is the definition of OTMANN distance. This formulation is general to many neural network architectures, is intuitive and can be solved efficiently using OT algorithms. Then the paper proposes to use EA to optimize the acquisition function and outperforms baselines on a variety of datasets. The paper is clearly written and could have big impact on NAS because the proposed method is not tailored for a specific search space and BO is a widely-used black-box optimization framework. I have two suggestions: 1) Since I do not have a good intuitive how expensive it is to optimize equation (3), it would be great to summarize the computation time (by percentage) of each component (distance calculation, GP regression, EA for acquisition function, etc.) in a table. 2) Unfortunately, this paper did not achieve state-of-the-art result for CIFAR 10. Actually, its accuracy is far from impressive. Although the paper argues that it uses less computational resources and a more general search space, I am still wondering whether it is possible to achieve state-of-the-art accuracy and how much computation is needed to achieve that.

Reviewer 2



The paper describes a new distance metric for neural networks based on optimal transport. Based on this distance, the authors introduce a new Bayesian optimization strategy to optimize the architecture of neural networks. Overall I think the methods is interesting and that the proposed distance might be also of interest for other applications than Bayesian optimization. However, it seems that computing the distance also requires some manual tuning such as defining the cost matrix which in turn requires expert knowledge. My only point of criticism is that this might hinder its success in automated machine learning applications, where human interaction should be reduced to a minimum. Having said that, the method is well explained and the authors provide some meaningful empirical insights, such that the paper represents an important contribution for Bayesian optimization. The following points need some further clarification: 1) How many actual function evaluation did every method perform in Figure 2. i.e how many architectures were trained? In the figure it seems that every method returns something after a discrete time step? 2) What do you mean with at least 5 independent runs for each method in the caption of Figure 2? Why did you perform less runs for some methods? 3) How are pending jobs treated in the parallel set up described in section 5? 4) In table 3, it says CIFAR-10 has 1k features instead of 32*32*3=3072. I guess that this is a typo? --- Post Rebuttal --- I thank the authors for taking their time to write the rebuttal and clarifying my points.

Reviewer 3



This paper introduces a way of doing Bayesian optimization over neural network architectures. The core of the method is a new distance measure that the authors introduced to measure similarities between different architectures. Given this new distance measure, one could then define a 'kernel' which could be used in a traditional way to perform Bayesian optimization. This work is novel and quite impressive. The experimental results backs up the claim of the paper. The paper is also well-written. I, however, have a few questions for the authors: 1. How does the EA algorithm used in the paper compare to other evolution based methods like: https://arxiv.org/pdf/1802.01548.pdf. 2. Can the proposed approach be used in conjunction with population based training in order to reduce overall compute time. 3. How would a pre-defined distance function compare to a learned distance function by optimizing the marginal likelihood. Can this approach be combined with learning?